# *Tm*PGRP-SA regulates Antimicrobial Response to Bacteria and Fungi in the Fat Body and Gut of *Tenebrio molitor*

**DOI:** 10.3390/ijms21062113

**Published:** 2020-03-19

**Authors:** Maryam Keshavarz, Yong Hun Jo, Tariku Tesfaye Edosa, Young Min Bae, Yeon Soo Han

**Affiliations:** Department of Applied Biology, Institute of Environmentally-Friendly Agriculture (IEFA), College of Agriculture and Life Sciences, Chonnam National University, Gwangju 61186, Korea; mariakeshavarz1990@gmail.com (M.K.); bunchk.2000@gmail.com (T.T.E.); ugisaka@naver.com (Y.M.B.)

**Keywords:** *Tenebrio molitor*, *TmPGRP-SA*, nuclear factor κB, expression pattern, antimicrobial peptide

## Abstract

Antimicrobial immune response is mediated by a signal-transducing sensor, peptidoglycan recognition protein-SA (PGRP-SA), that can recognize non-self molecules. Although several studies have focused on the involvement of *Drosophila* PGRP-SA in antimicrobial peptide (AMP) expression in response to infections, studies on its role in *Tenebrio molitor* are lacking. Here, we present a functional analysis of *T. molitor* PGRP-SA (*Tm*PGRP-SA). In the absence of microbes, *TmPGRP-SA* was highly expressed in the late-larval fat body, followed by hemocytes, and gut. Interestingly, following *Escherichia coli*, *Staphylococcus aureus*, and *Candida albicans* infections, the mRNA level of *TmPGRP-SA* was significantly upregulated in both the fat body and gut. *TmPGRP-SA* silencing had a significant effect on the mortality rates for all the microbes tested. Moreover, *TmPGRP-SA* is required for regulating the expression of eight AMP genes namely *TmTenecin-1*, -*2*, and *-4*; *TmDefensin-1* and *-2*; *TmColeoptericin-1*; and *TmAttacin-1b* and *-2* in the fat body in response to *E. coli* and *S. aureus* infections. *TmPGRP-SA* is essential for the transcription of *TmTenecin-2*, *-4*; *TmDefensin-2*; *TmColeoptericin-1*, *-2*; and *TmAttacin-1a*, *-1b*, and *-2* in the gut upon *E. coli* and *C. albicans* infections. However, *TmPGRP-SA* does not regulate AMP expression in the hemocytes. Additionally, *Tm*Dorsal isoform X2, a downstream Toll transcription factor, was downregulated in *TmPGRP-SA*-silenced larval fat body following *E. coli* and *S. aureus* challenges, and in the gut following *E. coli* and *C. albicans* challenges.

## 1. Introduction

Innate immunity serves to limit a wide variety of molecules derived from infectious pathogens and neutralizes their virulence through germline-encoded pattern recognition receptors termed pathogen recognition receptors (PRRs). PRRs expressed on host immune cells detect highly conserved microbial surface-derived molecules referred to as pathogen-associated molecular patterns (PAMPs) that are vital for microbial fitness and survival [1,2,3].

Among all the microbe-derived immune elicitors, peptidoglycans (PGNs) are major virulence components of Gram-positive and Gram-negative bacteria, and are structurally divided into two groups, lysine (Lys) type and *meso*-diaminopimelic acid (DAP) type PGNs [4]. PGNs are unique and well-conserved targets that trigger innate immune responses by binding to members of the PGN recognition protein (PGRP) family in the host. Members of the PGRP family are classified into two types, PGRP-S (short) and PGRP-L (long) [5]. In insects as well as mammals, the short PGRPs and carboxyl-terminal of long PGRPs share conserved amino acids that are homologous to bacteriophage type 2 amidases and T7 lysozymes which are known as amidase type-2 domain or PGRP domain [6]. Early evolutionary separation of insect PGRPs resulted in their categorization into catalytic and non-catalytic PGRPs [7,8]. The former class shares a conserved three-amino-acid structure that cleaves the amide bond between N-acetylmuramic acid (MurNAc) and L-alanine (L-Ala) of PGNs [9,10]. The latter (sensors) bind to PGNs but do not hydrolyze PGNs due to a missing cysteine residue. Instead they activate the intracellular Toll and immune deficiency (IMD) signaling pathways and proteolytic cascades [11,12].

In *Drosophila*, both types of PGNs are recognized by the PGRPs to subsequently activate the nuclear factor kappa-light-chain-enhancer of activated B cells (NF-κB)-dependent signaling cascades. The Toll pathway is activated through recognition of lys-type PGNs which trigger the Toll/Dorsal signaling, whereas the IMD pathway is triggered through detection of DAP-type PGNs which promote the activation of IMD/Relish signaling, and ultimately results in antimicrobial peptides (AMPs) production [13,14]. In this context, 13 PGRP genes from *Drosophila* including *Dm*PGRP-SA, -SB1, -SB2, -SC1a, -SC1b, -SC2, and -SD (PGRP-S) and *Dm*PGRP-LA, -LB, -LC, -LD, -LE, and -LF (PGRP-L) have been characterized [5].

Phylogenetic studies have inferred that PGRP-S plausibly evolved or originated from a common ancestral gene comprising two introns, and that one (*Dm*PGRP-SB2) or both (*Dm*PGRP-SB1, -SC1a and 1b, -SC2, SD) introns were lost during evolution. In contrast to other PGRP-S, *Drosophila* PGRP-SA (*Dm*PGRP-SA; 20-kDa) retains both introns in the PGRP domain [15]. Furthermore, the presence of an amino-terminal signal peptide indicates that *Dm*PGRP-SA is an extracellular recognition (secreted) protein expressed in the hemolymph [5]. Further, almost all PGRP-S contain two conserved cysteine residues in close proximity that can be linked by disulfide bridges, which are necessary for the three-dimensional conformation of the PGRP domain. Strikingly, a transition mutation (guanine → adenine) in *Dm*PGRP-SA led to the substitution of one cysteine (Cys80 → Tyr80) that is conserved in <90% of the PGRPs, and affects Toll activation in response to infection with Gram-positive bacteria [16].

A complex of two PRRs, namely *Dm*PGRP-SA and Gram-negative binding protein-1 (*Dm*GNBP-1) is required for recognition of Gram-positive bacteria and for triggering the Toll signaling pathway [17,18,19]. *Dm*GNBP-1 hydrolyzes the PGN resulting in free reducing ends of MurNAc, which are further sensed by *Dm*PGRP-SA [20]. It was previously assumed that *Dm*PGRP-SA lacked enzymatic function and served merely to sense the Lys- and DAP-type PGNs upstream of the intracellular Toll pathway. However, further studies revealed that *Dm*PGRP-SA has an intrinsic L,D-carboxypeptidase activity towards the DAP-type PGNs due to Ser158 and His15 residues [21,22]. Consistent with this finding, recent studies on PGRP-SA from *Bombus ignitus*, *Apis mellifera*, and *Megachile rotundata* revealed preferential binding of PGRP-SA to DAP-type PGNs than to Lys-type PGNs [23].

The mealworm beetle, *Tenebrio molitor*, presents some advantages when compared to the other invertebrate models (e.g., *D. melanogaster*) with respect to studying pathogenic infections and immune responses. For instance, the heat tolerance feature of the insect enables it to be maintained at 37 °C, which is an environmental cue for the expression of virulence factors in many pathogens. This makes *T. molitor* a great model to study host-pathogen interactions [24,25].

In the past decade, several extracellular and intracellular events of the *T. molitor* Toll signaling pathway have been investigated. In *T. molitor*, the Lys-type PGNs, DAP-type PGNs, and β-1, 3-glucan are sensed by the *Tm*PGRP-SA/*Tm*GNBP-1 complex and the Gram-negative binding protein 3 (*Tm*GNBP3), respectively, following which they trigger proteolytic cascades which lead to the extracellular cleavage of the Toll endogenous ligand spätzle (*Tm*Spz). The active form of *Tm*Spz binds to the ectodomain of the transmembrane Toll receptors (e.g., *TmToll-like receptor 7* [*TmToll-7*]) resulting in Toll-dependent AMP production [26,27,28,29]. In *T. molitor*, nine *Tm*Spz ligands including *TmSpz-1b*, *-3*, -*4*, -*5*, -*6,* -*7*, -*7a*, -*7b*, *7b,* and *-like* have been identified (unpublished data) [30]. However, the effect of *Tm*PGRP-SA expression on each of the *Tm*Spz genes remains unclear. Concordantly, the intracellular protein cassettes of the Toll pathway comprising *Tm*MyD88 [31], Pelle, and Tube are essential for transducing the signal to *Tm*Cactin [32], a positive regulator that interacts with *Tm*Cactus, for ensuring the nuclear translocation of *Tm*DorX2 [33] and robust transcription of the *Tm*AMP gene. In agreement with this, earlier immune studies in other invertebrates such as *D. melanogaster*, *Manduca sexta*, and *Marsupenaeus japonicus* have revealed the same mechanism of indirect Toll activation [22,34,35].

In vitro and in vivo studies have primarily addressed the role of *Tm*PGRP-SA as an innate immune recognition molecule that initiates the prophenoloxidase (proPO) cascade as well as the Toll signaling pathway [27,36]. However, the tissue-specific role of *Tm*PGRP-SA in inhibiting the Toll pathway needs to be elucidated. In this study, we sought to understand the functional role of extracellular *Tm*PGRP-SA in the survival and AMP gene expression of *T. molitor* larvae in response to *E. coli*, *S. aureus*, and *C. albicans* challenges using RNA interference (RNAi). Furthermore, we analyzed the expression pattern of NF-κB genes in *T. molitor* larvae following *Tm*PGRP-SA silencing and infection with the aforesaid microbes.

## 2. Results

### 2.1. Gene Organization, cDNA Analysis, and Phylogenetic Tree of TmPGRP-SA

The full-length open reading frame (ORF) of *Tm*PGRP-SA consists of 594 base pairs (bps), and encodes a polypeptide of 197 amino acids (aa). SignalP-5.0 analysis showed that *Tm*PGRP-SA contains a signal peptide at the N-terminal and with cleavage site predicted between the amino acid residues at position 19 and 20. The deduced *Tm*PGRP-SA protein contains a N-acetylmuramoyl-L-alanine amidase/PGRP domain (C_29_ to R_193_), based on the predictions from InterProScan 5.0 and BLASTx analyses (Figure 1). SWISS-MODEL and PyMOL visualized the amidase catalytic site (active site) and chemical (substrate) binding site (Appendix A).

Multiple sequence alignment of *Tm*PGRP-SA and other representative insect proteins revealed the relationship between the residues in the PGRP domains of corresponding sequences. Notably, the most conserved amino acids (displayed with asterisk) include proline, leucine, glycine, cysteine, valine, serine, and tryptophan (Figure 2A). The percent identity based on the full-length ORF of *Tm*PGRP-SA and that from other insects indicated that *Tm*PGRP-SA was highly similar to PGRP-SA from *Tribolium castaneum* (76% similarity), followed by 50% and 43% identity with the Orthopterans, *Locusta migratoria* (*Lm*PGEP-SA) and *Gryllus bimaculatus* (*Gb*PGRP-SA), respectively. Furthermore, a minimum and maximum similarity of *Tm*PGRP-SA with Hymenoptera (37–38%) and Diptera (43–45%) orders was also noted, suggesting that *Tm*PGRP-SA shares higher similarity with Diptera than Hymenoptera. The least similarity was observed with the order Lepidoptera (27%) (Appendix A).

An ML tree was constructed based on the protein sequences of PGRP-SA from twelve representative insect species and one human homolog (outgroup) (Figure 2B). The phylogenetic tree showed that the PGRP-SA isoforms from *T. molitor* and *T. castaneum* clustered together, and that *Tm*PGRP-SA had a close evolutionary position with respect to *Lm*PGEP-SA and *Gb*PGRP-SA (Orthoptera). Additionally, all Diptera and Hymenoptera species were grouped into two independent clusters. Phylogenetic analysis of the selected Diptera PGRP-SA protein sequences depicted two separate clusters, one formed by *Drosophila* (*Dm*PGRP-SA (*D. melanogaster*), *Ds*PGRP-SA (*D. simulans*), *Db*PGRP-SA (*D. busckii*)) and another by other flies (*BlPGRP-SA* (*B. latifrons*), *Zc*PGRP-SA (*Z. cucurbitae*)) (Figure 2B).

### 2.2. Developmental and Tissue Distribution of TmPGRP-SA

Insect PGRP-Ss (e.g., *Drosophila*) were previously shown to exhibit a tissue-specific expression profile, with short PGRPs being expressed almost exclusively in the fat body cells, with some expression also being detectable in the epidermal cells, gut, hemolymph, and cuticle [5]. To determine if *TmPGRP-SA* was active during developmental stages and in the larval or adult tissues, we sought to evaluate its development- and tissue-specific expression using RT-qPCR (Figure 3). Notably, *TmPGRP-SA* mRNA was detected at all the developmental stages tested. Although *TmPGRP-SA* expression was hardly detectable in the larval and pupal stages, its expression was highly increased in the adult stages, with the highest expression in the 1-day-old adult, followed by a rapid decline in expression in the older adults (Figure 3A). Further, we found that *TmPGRP-SA* mRNA was detectable in all the tissues analyzed. *TmPGRP-SA* showed elevated expression in the larval fat body, followed by hemocytes, gut, and Malpighian tubules. Additionally, we observed the lowest level of *TmPGRP-SA* mRNA expression in the integument (Figure 3B). RT-qPCR analysis of adult tissues revealed a markedly different pattern of *TmPGRP-SA* transcript expression in all tissues, with the highest expression detected in the integument and fat body, followed by ovary and Malpighian tubules. *TmPGRP-SA* mRNA was barely detectable in the adult hemocytes, gut, and testis (Figure 3C).

### 2.3. TmPGRP-SA is Upregulated following Microbial Infection in vivo

Previous studies have demonstrated that in *D. melanogaster*, PGRP-SA is involved in Toll-dependent immune defense against Gram-positive bacteria but not against fungal or Gram-negative bacterial infections [17]. Therefore, to investigate the involvement of *Tm*PGRP-SA in promptly detecting various infectious pathogens, we examined the response of *T. molitor* larvae (whole body and multiple tissues) to infection with *E. coli*, *S. aureus*, and *C. albicans* at specific time points (3, 6, 9, 12, and 24 h post-challenge) (Figure 4). We observed significantly elevated levels of *TmPGRP-SA* mRNA when *E. coli* and *S. aureus* were injected in the whole body of *T. molitor* larvae (Figure 4A,B). However, *TmPGRP-SA* expression showed a slight but significant induction at 3 h (or no response) to the *C. albicans* at the other time points (*p <* 0.05) (Figure 4C). Upon bacterial infection, a gradual increase in the transcript levels of *TmPGRP-SA* leading to a 40-fold upregulation in mRNA expression was noted with respect to the PBS-injected control at 24 h post-infection (Figure 4A,B). In the larval fat body of *T. molitor*, infection-mediated induction of *TmPGRP-SA* was significantly higher than in the PBS-injected cohorts (*p* < 0.05) (Figure 4A–C). *E. coli* and *S. aureus* challenge moderately increased *TmPGRP-SA* expression in the fat body, with the highest level observed at 24 h (Figure 4A,B). In the gut, induction of *TmPGRP-SA* mRNA by *E. coli* was stronger than in response to *S. aureus* and *C. albicans* (Figure 4A–C). Following microbial infections, induction of *TmPGRP-SA* in the hemocytes varied depending on the type of microbe. Whereas challenge with *C. albicans* did not induce *TmPGRP-SA* expression relative to that observed in PBS-injected cohorts (Figure 4C); exposure to the Gram-negative and Gram-positive bacteria, *E. coli* and *S. aureus*, respectively, triggered significant upregulation in *TmPGRP-SA* at the early time points (6 h), but did not persist at a high level at 12 and 24 h post-challenge (Figure 4A,B). Additionally, in agreement with the aforementioned results in hemocytes, high accumulation of *Tm*PGRP-SA in the hemolymph of *T. molitor* following challenge with lys-type and DAP-type PGNs was previously observed [37]. These findings highlight the role of a bacterial (but not fungal) elicitor, and that of *Tm*PGRP-SA as a sensor in the fat body, gut, and hemocytes of *T. molitor* larvae against *E. coli* and *S. aureus* challenges.

### 2.4. TmPGRP-SA Silenced Larvae are Susceptible to Microbial Infections

The *Drosophila* Toll pathway can be activated by Gram-positive Lys-type and Gram-negative DAP-type PGNs that are recognized by *Dm*PGRP-SA [22]. As *Dm*PGRP-SA-silenced flies are more susceptible to Gram-positive bacteria than to Gram-negative bacteria and fungi [17], we investigated whether suppression of *TmPGRP-SA* followed by *E. coli*, *S. aureus*, and *C. albicans* infections would lead to increased mortality of *T. molitor* larvae in a 10-day period. As shown Figure 5A, *TmPGRP-SA* was effectively silenced (85% reduction) on the second day post the dsRNA treatment. The percent survival of *TmPGRP-SA*-silenced larvae after challenges with *E. coli* (30%, Figure 5B) and *C. albicans* (20%, Figure 5D) were significantly compromised compared to that of the control larvae (70% and 60% respectively; *p* < 0.05). *TmPGRP-SA* knockdown larvae survived up to 50% compared to 70% survival of ds*EGFP*-treated control following *S. aureus* challenge (Figure 5C). These data suggest the prominent role of *TmPGRP-SA* in activation of immunity against microbial infections. Simultaneously, we assessed the expression of Toll-dependent genes (AMP and NF-κB genes) in *TmPGRP-SA*-silenced larvae following the microbial challenges.

### 2.5. Induction of AMPs is Regulated by TmPGRP-SA in Immunocompetent Tissues

We hypothesized that the survival phenotypes of insects challenged with different microbes may be due to differential expression of AMP genes. To investigate the downstream signal transduction of *TmPGRP-SA* following infection which leads to NF-κB translocation and induction of immune responses, we determined the expression of 14 AMP genes in larval fat body, gut, and hemocytes of ds *TmPGRP-SA*- and ds*EGFP*-treated groups 24 h post-*E. coli*, -*S. aureus*, and -*C. albicans* challenges.

Infection of ds*EGFP*-treated larvae with *E. coli* induced the expression of *TmTene1* (Figure 6A), *TmTene2* (Figure 6B), *TmTene4* (Figure 6D), *TmAtt1a* (Figure 6E), *TmAtt1b* (Figure 6F), *TmAtt2* (Figure 6G), *TmCole1* (Figure 6H), *TmDef1* (Figure 6J), *TmDef2* (Figure 6K), and *TmCec2* (Figure 6L) genes in the fat body. It should be noted that a significant downregulation of all these AMPs was observed in *E. coli*-infected larvae in which *TmPGRP-SA* was silenced (*p* < 0.05) (Figure 6). Exposing *TmPGRP-SA*-depleted cohorts to *S. aureus* markedly decreased the expression of nine AMP genes, namely *TmTene1* (Figure 6A), *TmTene2* (Figure 6B), *TmTene3* (Figure 6C), *TmTene4* (Figure 6D), *TmAtt1b* (Figure 6F), *TmAtt2* (Figure 6G), *TmCole1* (Figure 6H), *TmCole2* (Figure 6I), *TmDef1* (Figure 6J), and *TmDef2* (Figure 6K) relative to that in the ds*EGFP*-treated groups. In contrast, impairment of *TmPGRP-SA* expression did not affect *C. albicans*-induced expression of *TmTene1* (Figure 6A), *TmTene4* (Figure 6D), *TmAtt1b* (Figure 6F), *TmCole2* (Figure 6I), and *TmCec2* (Figure 6L). However, *TmPGRP-SA* silencing slightly downregulated few AMP mRNAs including *TmTene2* (Figure 6B), *TmAtt2* (Figure 6G), *TmCole1* (Figure 6H), *TmDef1* (Figure 6J), and *TmTLP1* (Figure 6M) in the larval fat body following *C. albicans* challenge.

In the gut of the control (ds*EGFP*-treated) larvae, *E. coli* triggered a potent transcription of *TmTene2* (Figure 7B), *TmTene4* (Figure 7D), *TmAtt1a* (Figure 7E), *TmAtt1b* (Figure 7F), *TmAtt2* (Figure 7G), *TmCole1* (Figure 7H), *TmCole2* (Figure 7I), and *TmDef2* (Figure 7K). However, the gut of ds*TmPGRP-SA*-silenced larvae showed lower expression of all AMPs except *TmTene1* (Figure 7A), *TmDef1* (Figure 7J), *TmCec2* (Figure 7L), and *TmTLP1* (Figure 7M) following challenge with *E. coli*. In contrast, when infected with *S. aureus*, induction of almost all AMPs including *TmTene1* (Figure 7A), *TmTene3* (Figure 7C), *TmTene4* (Figure 7D), *TmAtt1a* (Figure 7E), *TmAtt1b* (Figure 7F), *TmCole1* (Figure 7H), *TmDef1* (Figure 7J), *TmDef2* (Figure 7K), *TmCec2* (Figure 7L), *TmTLP1* (Figure 7M), and *TmTLP2* (Figure 7N) was significantly higher than that of the control groups in the gut (*p* < 0.05). Interestingly, *C. albicans* challenge markedly induced the mRNA levels of 11 AMP genes in the gut of ds*EGFP*-treated control larvae, whereas these transcripts were decreased in ds*TmPGRP-SA*-treated larvae (Figure 7). These results indicated that the gut responds to infections and that immune responses to *E. coli* and *C. albicans* are conveyed by *TmPGRP-SA*.

Unlike lower expression of AMPs observed in the hemocytes following microbial challenges, antimicrobial responses were strong in both fat body and gut (Appendix A). In this context, no differences in *TmTene1* (Appendix A) and *TmAtt1a* (Appendix A) transcription were found between ds*EGFP*- and ds*TmPGRP-SA*-treated larvae against all microbes tested. On the other hand, a mild downregulation of *TmTene2* (Appendix A) was noticed following *TmPGRP-SA* silencing upon *E. coli*-, *S. aureus*-, and *C. albicans*-infection in the hemocytes. Altogether, *TmPGRP-SA* does not appear to affect AMP regulation in the hemocytes.

### 2.6. Effect of TmPGRP-SA Knockdown on NF-κB Gene Expression

The above data demonstrated that as a receptor *Tm*PGRP-SA plays an important role in the recognition of bacterial pathogens such as *E. coli*, *S. aureus*, and *C. albicans* through the induction of AMP gene expression. Thus, we asked whether the depletion of *TmPGRP-SA* would impair the mRNA expression of *T. molitor* transcription proteins, including *TmRelish*, *TmDorX1*, and *TmDorX2* in the fat body, gut, and hemocytes following *E. coli*, *S. aureus*, and *C. albicans* infections.

The effect of *TmPGRP-SA* knockdown on the induction of *TmDorX2* and *TmRelish*, encoding Toll- and IMD-pathway mediated transcription proteins, respectively, by *E. coli* and *C. albicans* showed a similar tendency in both fat body and gut (Figure 8). In *TmPGRP-SA*-silenced larvae, the expression of *TmDorX2* and *TmRelish* were significantly decreased in both the fat body and gut following *E. coli* and *C. albicans* challenges (Figure 8B,C) unlike in the case of *TmDorX1*, which was upregulated in the fat body in *TmPGRP-SA*-depleted group following challenges with all microbes (Figure 8A). Although, *TmDorX1* expression in the gut appeared to be downregulated following *E. coli* and *C. albicans* challenges (Figure 8A). Of note, expression of NF-κB genes were insensitive to regulation by *TmPGRP-SA* in the hemocytes (Figure 8A–C). Overall, our data highlight that loss of *TmPGRP-SA* caused significant downregulation of *TmDorX2* and *TmRelish* after *E. coli* and *C. albicans* infections in both the immune-related organs, fat body, and gut.

## 3. Discussion

In the present study, we identified the same PGRP-SA from *T. molitor* that was previously identified by Lee and his colleagues (accession number: AB219970.1), and it showed a high degree of high identity with *Drosophila* PGRP-SA [36]. Like all known PGRPs, *Tm*PGRP-SA possesses the same conserved domain and a disulfide bond between cysteine residues which is essential for protein stability. Remarkably, the *Dm*PGRP-SA contains a signal peptide (N-terminus) and a single PGRP domain. The cleaved PGRP-SA has a single PGRP domain, indicating that it is a secreted protein [5,21]. In agreement with *Drosophila* studies, the N-terminal residues of *Tm*PGRP-SA correspond to the putative signal peptide and contains one PGRP domain. Given that the PGRP domains of both insects and mammals share conserved amino acids that are homologous to N-acetylmuramoyl-L-alanine amidase (T7 lysozymes), we found only one tyrosine (Y) residue of three conserved amino acids (histidine, tyrosine, and lysine) in the active site of *Tm*PGRP-SA, suggesting that like in *Drosophila*, *Tm*PGRP-SA may not have amidase activity [6,36,38]. However, further studies in *Dm*PGRP-SA have revealed that a serine residue (Ser158) is a key amino acid for carboxypeptidase activity toward DAP-type PGNs and Toll activation, which is highly conserved in PGRPs that lack amidase catalytic activity [21]. Furthermore, PGRP-SA from *B. ignitus*, *A. mellifera*, and *M. rotundata* had stronger interaction with DAP-type PGNs than with lys-type PGNs [23]. Consistent with these findings, *Tm*PGRP-SA possesses the conserved serine residue in its active site (threonine174, serine175, serine176), suggesting that *Tm*PGRP-SA may have carboxypeptidase activity. Similar to PGRP-SA in *D. melanogaster*, *A. mellifera*, and *B. ignites*, our results show that *Tm*PGRP-SA has four conserved cysteine residues (Cys^29^, Cys^66^, Cys^72^, and Cys^151^) that may be linked by disulfide bonds. However, a crystal structure of *Tm*PGRP-SA is needed to elucidate the conformation [21,23,39]. Phylogenetic studies highlighted that Hymenoptera diverged from Diptera and Lepidoptera 300 million years ago and this divergence was after the evolution of Coleoptera [40]. Results from the present study show that Hymenoptera insects had fewer similarities with Coleopteran than Dipteran and this difference may be the result of the evolutionary process as the clade of Hymenoptera are more ancient than Diptera [23].

Studies over the past decade have demonstrated that insect short PGRPs are constitutively present in the hemolymph and cuticle proteins, and are expressed mainly in the fat body and gut, and to a lesser extent in the hemocytes [41,42]. In this context, *BiPGRP-S* was expressed at a high level in the fat body and relatively lower level in the epidermis in *B. ignitus* worker bees [43]. Furthermore, in the Chinese oak silkworm, *Antheraea pernyi*, the tissue distribution of *PGRP-SA* (*ApPGRP-SA*) was found widely in the different immune-related tissues including the fat body, midgut, epidermis, and hemocytes [44]. Consistent with previous studies, *Tm*PGRP-SA transcripts were highly expressed in the larval fat body, followed by hemocytes and gut.

Innate immune response is triggered by signal-transducing sensors, the peptidoglycan recognition proteins (PGRPs), that recognize non-self molecules, such as the lys-type PGNs of Gram-positive bacteria, DAP-type PGNs of Gram-negative bacteria, and β-1,3-glucans of fungi [45]. Invertebrate PGRPs, like those in some mammals, have shown multivalent binding properties to a variety of pathogenic molecules [46,47]. In the oyster, *Crassostrea gigas*, PGRPS2 displays high binding affinity to lipopolysaccharide (LPS), PGN, mannan, *E. coli*, *S. aureus*, and the fungus *Yarrowia lipolytica* [48], whereas, *Ap*PGRP-SA can detect *S. aureus*, *M. luteus*, *E. coli*, and *C. albicans* [46]. Our results showed that the transcriptional level of *TmPGRP-SA* was significantly upregulated following challenges with *E. coli*, *S. aureus*, and *C. albicans* in both fat body and gut, which was in agreement with expression level of *ApPGRP-SA* in response to *E. coli*, Gram-positive bacteria and fungi in the same tissues [44]. Likewise, *T. castaneum* PGRP-SA (*TcPGRP-SA*) mRNA expression was significantly increased following *E. coli* and *Micrococcus luteus* infections in 3-day-old pupae [49]. Contrary to *E. coli-* and *S. aureus-*infected larvae, *TmPGRP-SA* transcription was not affected by *C. albicans* infection in the larval hemocytes. This is consistent with high accumulation of *TmPGRP-SA* in the hemolymph of *T. molitor* against lys-type and DAP-type PGNs [37]. Taken together, bacterial challenges induce *TmPGRP-SA* in the larval fat body, gut, and hemocytes of *T. molitor*, while fungal infection (*C. albicans*) upregulates its expression only in the fat body and gut.

Monitoring of *TmPGRP-SA*-silenced larvae following infection with the microbes showed that *TmPGRP-SA* contributed to survival. Of note, survival of the ds*TmPGRP-SA*-treated larvae was significantly affected by infection with all the microbes, however they succumbed more to *E. coli* and *C. albicans* infection than to *S. aureus*. It is known that when challenged with non-self molecules derived from pathogens, immunocompetent tissues initiate complex mechanisms to synthesize AMPs and trigger innate immunity. Therefore, expression and regulation of the AMP genes in the fat body, gut, and hemocytes were analyzed.

Comparative studies conducted previously on the two main Toll pathway components revealed that the expression level of *TmTene1*, *TmTene4*, *TmDef1*, *TmDef2*, *TmCole1*, *TmAtt1b*, and *TmCec2* were significantly decreased in the fat body of *E. coli*-challenged *T. molitor* following silencing of *TmDorX2* and *TmCactin* [32,33]. Our results, similarly, showed significant downregulation of these genes in the fat body of ds*TmPGRP-SA*-treated larvae upon *E. coli* challenge. This result was also confirmed by measuring the fold-change of *TmDorX2* expression in the *TmPGRP-SA*-silenced larval fat body following *E. coli* infection. Notably, the induction of *TmAttacin* family is regulated by *TmPGRP-SA* and *TmDorX2* in the fat body of *E. coli*-infected larvae [33]. Furthermore, following infection with *S. aureus* the mRNA expression of six AMPs, namely *TmTene1*, *TmTene4*, *TmDef1*, *TmDef2*, *TmCole2*, and *TmAtt1b* was suppressed in the fat body of *TmPGRP-SA*- *TmDorX2*-, and *TmCactin-*silenced groups [32,33]. However, fat body of *TmPGRP-SA-* and *TmDorX2-*depleted larvae merely control *C. albicans-*induced transcription of *TmDef1* [33]. Moreover, we propose that *Tm*PGRP-SA is a positive regulator in the fat body, and controls the expression levels of *TmTene1*, *TmTene2*, *TmTene4*, *TmDef1*, *TmDef2*, *TmCole1*, *TmAtt1b*, and *TmAtt2* (8 AMPs) upon *E. coli* and *S. aureus* infections (Figure 9).

In the gut, silencing of *TmPGRP-SA* and *TmDorX2* following *E. coli* infection can suppress *TmTene2*, *TmTene4*, *TmDef2*, *TmCole1*, *TmCole2*, *TmAtt1a*, *TmAtt1b*, and *TmAtt2* transcriptions (8 AMPs); while *C. albicans* infection can impair *TmTene1*, *TmTene2*, *TmTene4*, *TmDef1*, *TmDef2*, *TmCole1*, *TmCole2*, *TmAtt1b*, *TmAtt2*, *TmCec2*, and *TmTLP1* expression (11 AMPs) [33]. In addition, knockdown of *TmPGRP-SA* and *TmDorX2* significantly downregulates *TmCole2* and *TmAtt2* response to *S. aureus* challenge [33]. Intriguingly, *E. coli-* and *C. albicans-*mediated induction of *TmDorX1*, *TmDorX2*, and *TmRelish* was regulated by *TmPGRP-SA* in the gut. Collectively, *TmPGRP-SA* plays a key role in regulating of AMP gene expression in the fat body in response to *E. coli* and *S. aureus*, whereas *TmPGRP-SA* is required for the transcription of AMPs in the gut in response to *E. coli* and *C. albicans* (Figure 9). We should also mention that *TmPGRP-SA* is not a main regulator of AMPs in the hemocytes. The above results demonstrate that the mortality observed in *T. molitor* larvae following *TmPGRP-SA* knockdown may be attributable to impairment or abolishment of *TmPGRP-SA*-mediated signal transduction and antimicrobial defense.

## 4. Materials and Methods

### 4.1. Rearing Stock of T. Molitor

*T. molitor* stocks were reared on an artificial diet at 27 ± 1 °C, 60 ± 5% relative humidity, and under conditions of darkness. Young larvae were fed an artificial diet consisting of 170 g wheat flour, 0.5 g chloramphenicol, 20 g roasted soy flour, 0.5 g sorbic acid, 0.5 mL propionic acid, 10 g soy protein, and 100 g wheat bran in 200 mL of distilled water, autoclaved at 121 °C for 20 min. Healthy and fed 10th–12th instar *T. molitor* larvae (approximately 2.4 cm) were used for all the experiments. The experimental units maintained in an insectary with artificial diet during all experiments.

### 4.2. Bioinformatics Analysis for Identification and Sequence Characterization of TmPGRP-SA

To identify *TmPGRP-SA*, a local-tblastn analysis was carried out using the protein sequence of *T. castaneum* PGRP-SA (*Tc*PGRP-SA) (GenBank: P_008192927.1) as the query. InterProScan 5.0 [50] and BLASTx [51] were performed to predict the functional domain of *Tm*PGRP-SA by using the deduced amino acid sequence as a template. SignalP-5.0 was used to predict the signal peptide [52]. SWISS-MODEL [53] and PyMOL [54] were used to illustrate the *Tm*PGRP-SA protein structure. To obtain the sequence similarity between *Tm*PGRP-SA protein and its orthologs, multiple protein sequences listed in Table 1 were aligned using ClustalX2 [55]. A maximum likelihood (ML) tree (JTT matrix-based model) [56] was generated based on the protein sequences with 1000 bootstrap replicates using MEGA 7.0 program [57]. *Homo sapiens* peptidoglycan recognition protein 3 (*Hs*PGRP3) was used as an outgroup in the phylogenetic studies.

### 4.3. Gene Expression Analysis of TmPGRP-SA in Multiple Tissues and in Different Developmental Stages

To determine the spatial expression profile of *TmPGRP-SA* during development, samples were collected from egg (EG), young larvae (YL; 10th–12th instar), late-instar larvae (LL; 19th–20th instar), pre-pupae (PP), 1 to 7-day-old pupae (P1–P7), and 1- to 5-day-old adults (*n* = 20 per each stage). In addition to *TmPGRP-SA* developmental profiling, the tissue-specific expression pattern of *TmPGRP-SA* mRNA was examined using multiple larval and adult tissues of *T. molitor* including the integument, fat body, hemocytes, gut, Malpighian tubules, ovary, and testis.

### 4.4. RNA Isolation and cDNA Synthesis

Guanidine thiocyanate-based RNA lysis buffer was used to extract the total RNA from the samples following the modified LogSpin RNA isolation method as described in a previous study [33,58]. First-strand cDNA was synthesized using 2 μg of total RNA and oligo-(dT)^12–18^ primers in a reaction volume of 20 μL. The reaction was incubated at 42 °C for 5 min. The resultant cDNA was added to the AccuPower^®^ RT PreMix (Bioneer, Daejeon, Korea) kit and then incubated at 72 °C for 1 h. The synthesized cDNAs were stored at −20 °C until use.

### 4.5. Quantitative Reverse-Transcription PCR (RT-qPCR) Analysis

cDNA was diluted (1:20 with DNase/RNase free water) to establish standard curves for determining empirically optimal template concentration and the primer efficiency. The RT-qPCR reaction mix (20 µL) included 10 µL AccuPower^®^ 2X GreenStar qPCR Master Mix (Bioneer, Daejeon, Korea), 3 µL of 1:20 diluted cDNA template, and 2 µL of designed primers (*TmPGRP-SA*-qPCR-Fw and *TmPGRP-SA*-qPCR-Rv) (Appendix A; Table 2). Specific primers against *TmPGRP-SA*, as well as against the *T. molitor* housekeeping gene (60 S ribosomal protein L27a), were designed using Primer 3.0 plus (http://www.bioinformatics.nl/cgi-bin/primer3plus/primer3plus.cgi); primer sequences are presented in Table 2. For each sample, duplicate reactions in a total volume of 20 µL were performed with the following thermal program: 95 °C for 5 min, followed by 40 cycles at 95 °C for 15 s and amplification at 60 °C for 30 s. the relative mRNA expression levels were evaluated using the comparative C_T_ method (2^−ΔΔ*C*T^ method) [59].

### 4.6. Microbial Strains

The following microorganisms were obtained from the American Type Culture Collection (ATCC) and used in this study: *Escherichia coli* strain K12, *Staphylococcus aureus* strain RN4220, and *Candida albicans* strain AUMC 13529. *E. coli* and *S. aureus* were grown aerobically in Luria-Bertani (LB) broth, while *C. albicans* was cultured in Sabouraud Dextrose broth at 37 °C overnight under continuous shaking. Overnight cultures were centrifuged at 3500 rpm for 15 min at room temperature (around 25 °C), and washed and diluted in phosphate-buffered saline (PBS; pH 7.0). Finally, the cell concentrations (based on OD_600_ measurements) were adjusted to 1 × 10^6^ cells/μL for *E. coli* and *S. aureus*, and 5 × 10^4^ cells/μL for *C. albicans*.

### 4.7. TmPGRP-SA mRNA Expression Following Microbial Challenges

To elucidate the effect of microbial strains on *TmPGRP-SA* mRNA expression in *T. molitor*, the 10th–12th instar larvae (*n* = 80 per time point) were divided into the following four subgroups: three groups of larvae (*n* = 20 per group) were challenged with 1 µL *E. coli* (1 × 10^6^ cells/µL), *S. aureus* (1 × 10^6^ cells/µL), and *C. albicans* (5 × 10^4^ cells/µL), respectively, while the remaining 20 larvae were injected with PBS as control. The experimental groups were maintained in the insectary under identical rearing conditions and then the immune tissues were dissected at 3, 6, 9, 12, and 24 h post-injection. The fat body, hemocytes, and gut were collected for subsequent RNA extraction, cDNA synthesis, and RT-qPCR analyses as described above.

### 4.8. Preparation of Double-Stranded TmPGRP-SA

The SnapDragon-Long dsRNA design software (https://www.flyrnai.org/cgi-bin/RNAi_find_primers.pl) was used to design gene-specific primers (Table 2). For *TmPGRP-SA*, we synthesized double-stranded RNA (dsRNA) designing forward and reverse primers; the T7 promoter sequence was added to their 5’ ends. Primers (ds*Tm*PGRP-SA_Fw and ds*Tm*PGRP-SA_Rv) were used to amplify 316 bp amplicons from *T. molitor* cDNA using AccuPower^®^ Pfu PCR PreMix under the following cycling conditions: 95 °C for 2 min, 30 cycles of 95 °C for 20 s, 56 °C for 30 s, and final extension at 72 °C for 5 min (Appendix A; Table 2). For control, the DNA template was cloned from the *enhanced green fluorescent protein* (*EGFP*) gene present within the EGFP-C1 vector, and tailed with T7 promotor using ds*EGFP*_Fw and ds*EGFP*_Rv primers (Table 2). The amplicons were purified with AccuPrep^®^ PCR Purification Kit (Bioneer). Next, the dsRNA was transcribed in vitro using the EZ^TM^ T7 High Yield in vitro Transcription Kit (Enzynomics, Deajeon, Korea), as per the manufacturer’s protocol.

### 4.9. T. molitor Survival Bioassay

Survival bioassays were performed by silencing *TmPGRP-SA* in young larvae of *T. molitor* followed by microbial infections. A volume of 1 μL (1 μg) of the gene of interest and control, synthesized as previously described, was injected into 10th–12th instar larvae. At least 10–15 larvae were injected with ds*TmPGRP-SA* and ds*EGFP* per treatment, and the experiments were repeated three times to obtain a total of 45 RNAi-injected larvae per group. Then, three surviving larvae per group were used to confirm the knockdown efficiency of the target gene on the second day after injection. Subsequently, dsRNA-injected larvae (*n* = 10 per group) were infected with *E. coli* (1 × 10^6^ cells/larva), *S. aureus* (1 × 10^6^ cells/larva), and *C. albicans* (5 × 10^4^ cells/larva), and survivors were monitored every day for a 10-day period.

### 4.10. Analysis of the Effect of TmPGRP-SA Silencing on AMP and NF-κB Gene Expression Post-Microbial Challenge

The survival analyses conducted above underline the importance of *TmPGRP-SA* in *T. molitor* fitness. These assay results also raise the question about the cause of the reduced lifespan observed in *TmPGRP-SA* knockdown larvae infected with microbes. It is plausible that low expression of AMP genes could lead to the notable mortality rates. To test this hypothesis, we examined the gene expression profiles of 14 AMPs, including *TmTenecin*-1 (*TmTene1*), *TmTenecin*-2 (*TmTene2*), *TmTenecin*-3 (*TmTene3*), *TmTenecin*-*4* (*TmTene4*), *TmAttacin-1a* (*TmAtt1a*), *TmAttacin-1b* (*TmAtt1b*), *TmAttacin-2* (*TmAtt2*), *TmDefensin*-1 (*TmDef1*), *TmDefensin*-2 (*TmDef2*), *TmColeoptericin-1* (*TmCole1*), *TmColeoptericin-2* (*TmCole2*), *TmCecropin-2* (*TmCec2*), *TmThaumatin-like protein-1* (*TmTLP1*), and *TmThaumatin-like protein-2* (*TmTLP2*) in the *TmPGRP-SA*-silenced larvae following the microbial challenges. Furthermore, the fold-changes in mRNA of three *T. molitor* transcription factors, namely *TmRelish*, *TmDorsal X1* isoform (*TmDorX1*), and *TmDorsal X2* isoform (*TmDorX2*) were also evaluated (Table 2). Ds*EGFP* was used as a negative control, and PBS served as a wound control. Sample tissues (fat body, gut, and hemocytes) were collected 24 h post-challenge and then processed for cDNA synthesis, and RT-qPCR analysis using AMP-specific primers (Table 2).

### 4.11. Statistical Analysis

Three independent biological replicates were used for all experiments. Values were reported as mean ± standard error (SE). Differences between groups were analyzed using one-way statistical analysis of variance (ANOVA) and Tukey’s test; *p* values < 0.05 were considered significant. The results of the mortality assay were analyzed using the Kaplan-Meier plot (log-rank Chi-square test) in Excel (http://www.real-statistics.com/survival-analysis/kaplan-meier-procedure/real-statistics-kaplan-meier/).

## 5. Conclusions

*T. molitor* peptidoglycan recognition protein-SA (*Tm*PGRP-SA) has multivalent binding properties to Gram-negative and Gram-positive bacteria (*E. coli* and *S. aureus*), and fungi (*C. albicans*). *Tm*PGRP-SA is a positive regulator in the fat body and gut, and regulates the expression of eight of fourteen AMP genes. Invading microbes are sensed by *Tm*PGRP-SA which then likely transduces the signal to *Tm*DorX2, which as a transcription factor to initiate the antimicrobial defense response.

## Figures and Tables

**Figure 1 ijms-21-02113-f001:**
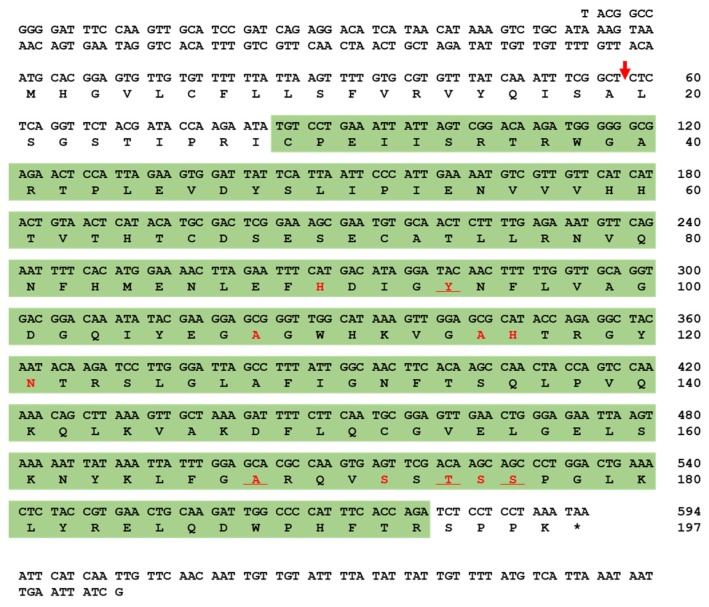
Nucleotide and deduced amino acid sequence of *T. molitor* peptidoglycan recognition protein-SA (*Tm*PGRP-SA). The nucleotide and amino acid sequence numbers are shown at the right margin, indicating that full-length open reading frame (ORF) sequence of *Tm*PGRP-SA consists of 594 nucleotides encoding 197 amino acid residues. The predicted signal peptide cleavage site is marked with a red arrow. The amidase/PGRP domain predicted by InterProScan 5.0 is highlighted in the green box. The amidase catalytic site (active site) is underlined and chemical (substrate) binding site is shown in red.

**Figure 2 ijms-21-02113-f002:**
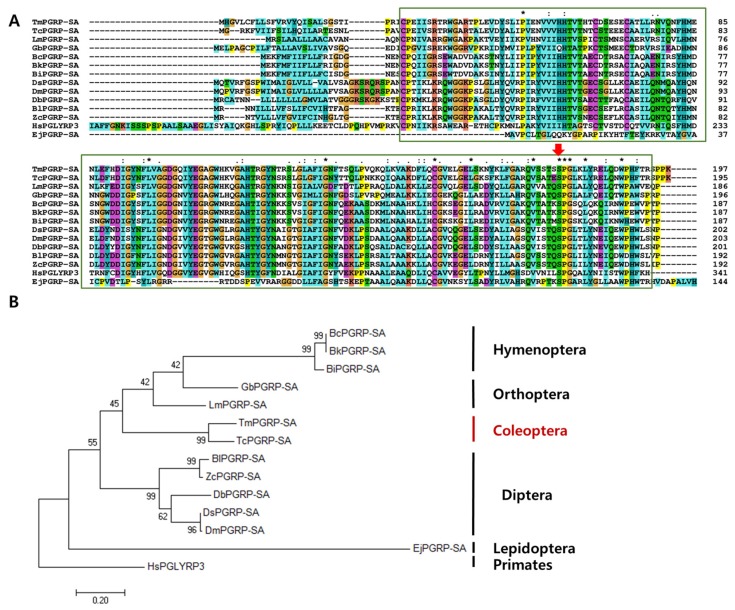
Phylogenetic analyses and alignment of *Tm*PGRP-SA amino acid sequence with those from other species. The amidase/PGRP domain is marked in green boxes. Asterisk, colon, and period symbols indicate conservation scores between the sequences of representative PGRP-SA proteins according to the Gonnet PAM 250 matrix (‘*’ > ‘:’ > ‘.’) and ‘-’ indicates internal or terminal gaps. The Red Green Blue (RGB) colors are identified based on RGB values (from 0 to 255). Red color shows 229, 51, and 25 RGB values, blue (25, 127, and 229), green (25, 204, and 25), cyan (25, 178, and 178), pink (299, 127, and 127), magenta (204, 76, and 204), yellow (204, 204, and 0), and orange (229, 153, and 76) (**A**). Red arrow indicates the conserved serine amino acids. Phylogenetic tree constructed using maximum likelihood (ML) method based on the Jones-Taylor-Thornton (JTT) matrix model in MEGA 7.0 (1000 bootstrap replicates). The MT tree were rooted with the protein sequences of *Tm*PGRP-SA (*Tenebrio molitor* peptidoglycan recognition protein SA), *Tc*PGRP-2 (*T. castaneum* REDICTED: peptidoglycan recognition protein 2; P_008192927.1), *Lm*PGRP-SA (*Locusta migratoria* peptidoglycan recognition protein SA; AFD54029.1), *Bc*PGRP-SA (*Bombus consobrinus* peptidoglycan recognition protein SA; ATL64828.1), *Bk*PGRP-SA (*Bombus koreanus* peptidoglycan recognition protein SA; ATL64813.1), *Gb*PGRP-SA (*Gryllus bimaculatus* peptidoglycan recognition protein SA; BBG28438.1), *Ds*PGRP-SA (*Drosophila simulans* peptidoglycan recognition protein SA; XP_002106687.1), *Db*PGRP-SA (*Drosophila busckii* peptidoglycan recognition protein SA; XP_002106687.1), *Ej*PGRP-SA (*Eumeta japonica* peptidoglycan recognition protein SA; GBP17419.1), *Dm*PGRP-SA (*Drosophila melanogaster* peptidoglycan recognition protein SA; CAD89124.1), *Bl*PGRP-SA (*Bactrocera latifrons* peptidoglycan recognition protein SA; JAI23539.1), *Zc*PGRP-SA (*Zeugodacus cucurbitae* peptidoglycan recognition protein SA; JAD13283.1), *Bi*PGRP-SA (*Bombus ignites* peptidoglycan recognition protein SA; ATL64812.1). *Hs*PGLYRP3 (*Homo sapiens* peptidoglycan recognition protein 3; AAI28116.1) was used as an outgroup (**B**).

**Figure 3 ijms-21-02113-f003:**
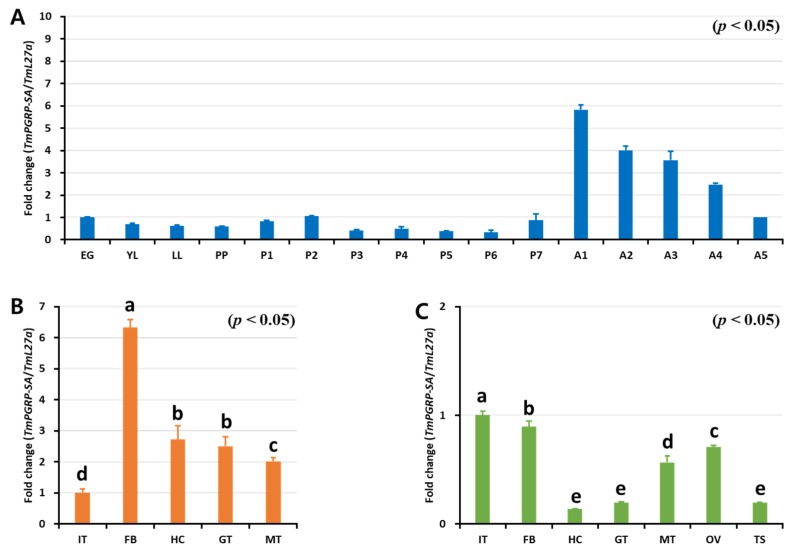
Expression of *TmPGRP-SA* in different developmental stages and multiple tissues of late-instar larvae and 5-day-old adults. RT-qPCR transcript analysis of *TmPGRP-SA* at different *T. molitor* developmental stages. EG: eggs, YL: young larvae, LL: late-instar larvae, PP: Pre-pupa, P1 – P7: 1 to 7-day-old pupa, and A1–A5: 1 to 5-day-old adults (**A**). mRNA profile of *TmPGRP-SA* in late-instar larval tissues (IT: integument, FB: fat body, HC: hemocytes, GT: gut, and MT: Malpighian tubules) (**B**) and in 5-day-old adult tissues (OV: ovary and TS: testis) using RT-qPCR (**C**). The results were normalized to *T. molitor* 60S ribosomal protein L27a (*TmL27a*). The mean values and SE were obtained from 20 insects per group. Bars in each graph with the same letter are not significantly different from each other (*p* > 0.05).

**Figure 4 ijms-21-02113-f004:**
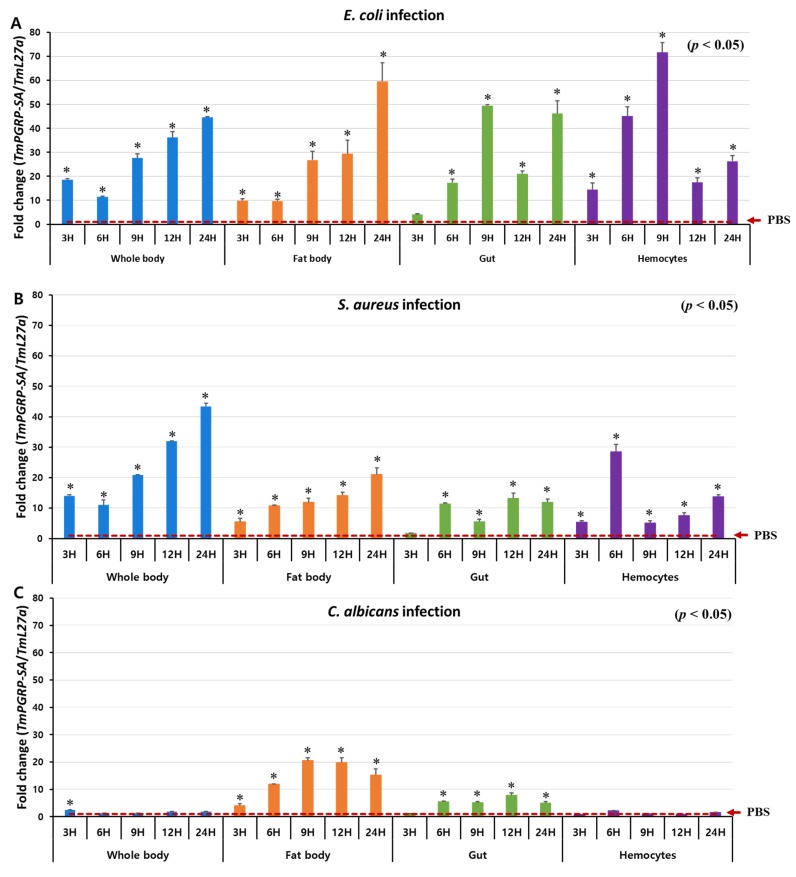
*TmPGRP-SA* is differentially induced following microbial challenges. Relative changes in gene expression (mean ± SE) of *TmPGRP-SA* in the whole body, fat body, gut, and hemocytes of *T. molitor* (10th–12th instar) larvae challenged with *E. coli* (**A**), *S. aureus* (**B**), and *C. albicans* (**C**) at the indicated time points by RT-qPCR (*n* = 20 per treatment group per time point). *TmPGRP-SA* mRNA expression was normalized to the reference gene, *T. molitor* 60S ribosomal protein L27a (*TmL27a*), followed by normalization to the PBS-injected control mRNA expression. Statistical significance is denoted with asterisks (*p* < 0.05).

**Figure 5 ijms-21-02113-f005:**
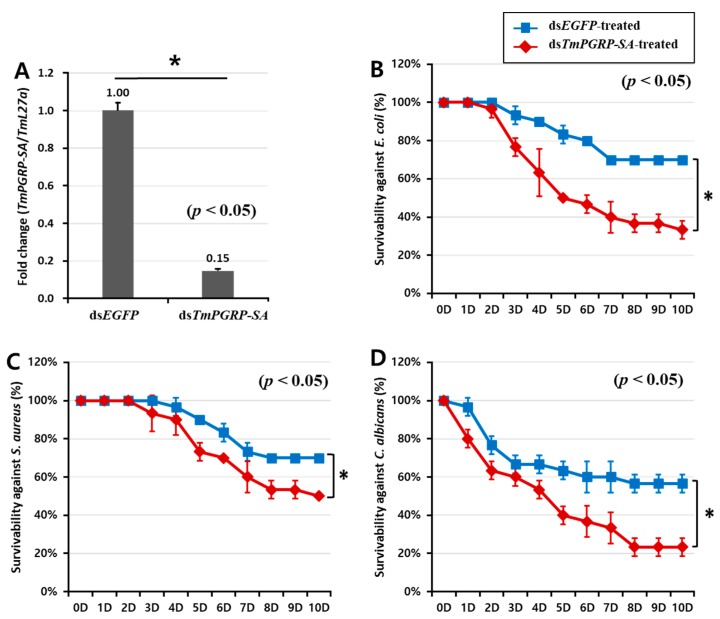
The pivotal role of *TmPGRP-SA* in the survival of *T. molitor* larvae following microbial infections. Knockdown efficiency of *TmPGRP-SA* in *T. molitor* larvae (*n* = 3 per group) quantified by mRNA transcription after *TmPGRP-SA* depletion compared to ds*EGFP* controls carried out on the second day post-injection (80%) (**A**). Lifespan curves of ds*EGFP*- and ds*TmPGRP-SA*-injected larvae (*n* = 10 per group) following immune challenge with *E. coli* (**B**), *S. aureus* (**C**), and *C. albicans* (**D**) over a ten-day period. Results presented are representative of three independent experiments. Asterisks depict significant differences between negative control and *TmPGRP-SA*-silenced larvae (*p* < 0.05).

**Figure 6 ijms-21-02113-f006:**
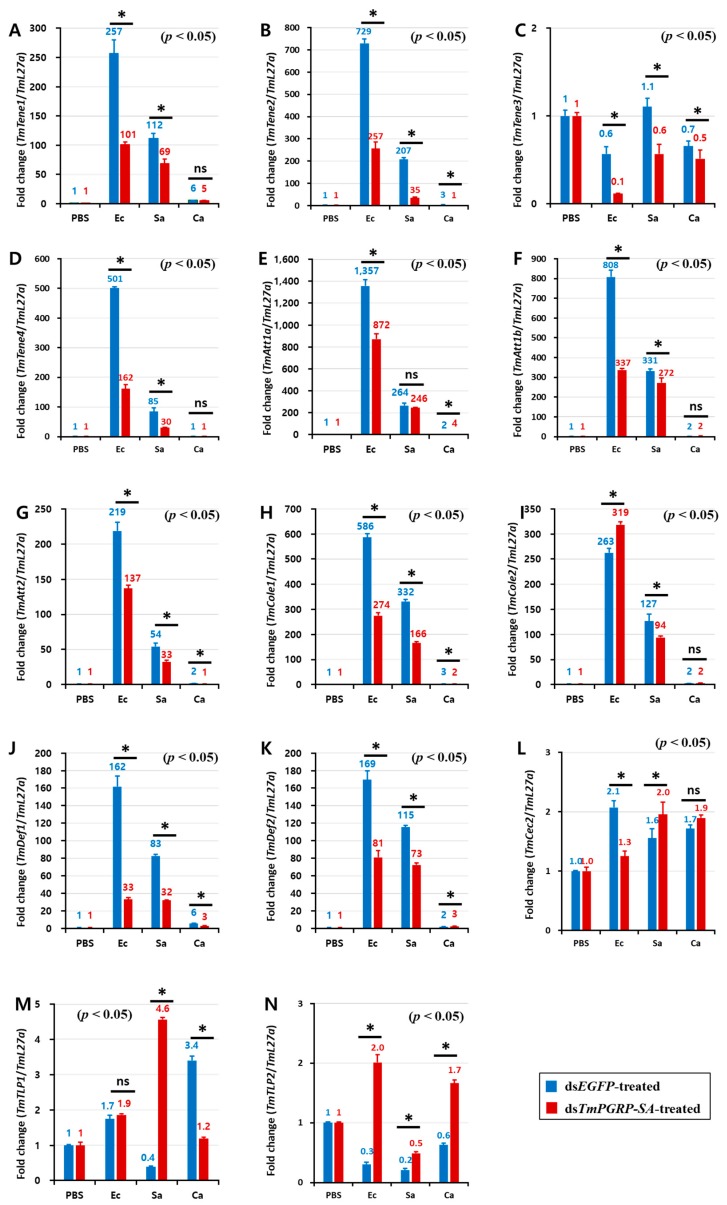
*TmPGRP-SA*-mediated transcription induction of antimicrobial peptides is lower in the fat body of ds*EGFP*-injected larvae following bacterial infections. Quantitative RT-PCR was used to measure the expression of fourteen antimicrobial peptide (AMP) genes including *TmTenecin-1* (*TmTene1,*
**A**); *TmTenecin-2* (*TmTene2,*
**B**); *TmTenecin-3* (*TmTene3,*
**C**); *TmTenecin-4* (*TmTene4,*
**D**); *TmAttacin-1a* (*TmAtt1a,*
**E**); *TmAttacin-1b* (*TmAtt1b,*
**F**); *TmAttacin-2* (*TmAtt2,*
**G**); *TmColeptericin-1* (*TmCole1,*
**H**); *TmColeptericin-2* (*TmCole2,*
**I**); *TmDefensin-1* (*TmDef1*, **J**); *TmDefensin-2* (*TmDef2,*
**K**); *TmCecropin-2* (*TmCec2,*
**L**); *TmTLP-1* (*TmTLP1,*
**M**); and *TmTLP-2* (*TmTLP2,*
**N**) in the *TmPGRP-SA-*silenced *T. molitor* larval fat body (*n* = 20 per group) in response to *E. coli* (Ec)*, S. aureus* (Sa)*,* and *C. albicans* (Ca) challenges. *EGFP* dsRNA injection served as a negative control, and the mRNA level of the respective AMP genes are presented relative to those for *TmL27a* as an internal control. Significant differences between ds*EGFP*- and ds*TmPGRP-SA*-treated groups are shown with asterisks (*p* < 0.05); ns = not significant.

**Figure 7 ijms-21-02113-f007:**
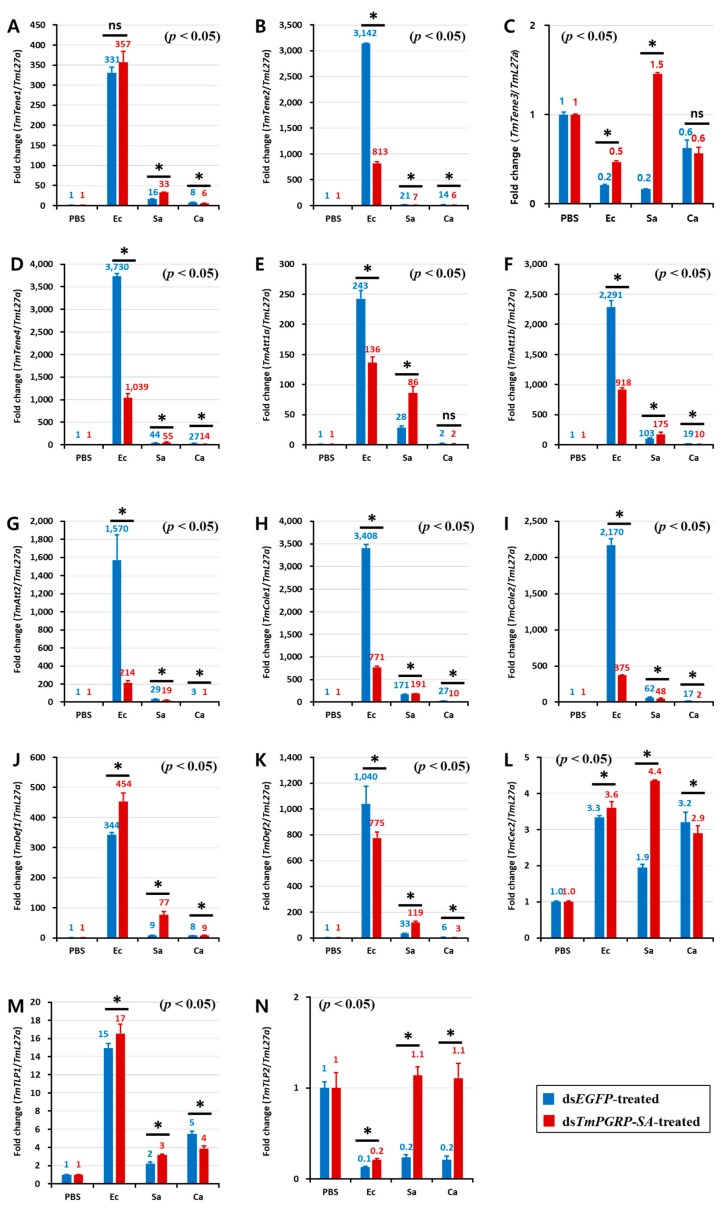
Expression of *TmPGRP-SA* is required for the induction of antimicrobial peptides in the gut. Effect of *TmPGRP-SA* reduction mediated by double-stranded RNA on infection-dependent expression of fourteen AMPs namely *TmTenecin-1* (*TmTene1,*
**A**), *TmTenecin-2* (*TmTene2,*
**B**), *TmTenecin-3* (*TmTene3,*
**C**), *TmTenecin-4* (*TmTene4,*
**D**), *TmAttacin-1a* (*TmAtt1a,*
**E**), *TmAttacin-1b* (*TmAtt1b,*
**F**), *TmAttacin-2* (*TmAtt2,*
**G**), *TmColeptericin-1* (*TmCole1,*
**H**), *TmColeptericin-2* (*TmCole2,*
**I**), *TmDefensin-1* (*TmDef1*, **J**), *TmDefensin-2* (*TmDef2,*
**K**), *TmCecropin-2* (*TmCec2,*
**L**), *TmTLP-1* (*TmTLP1,*
**M**), and *TmTLP-2* (*TmTLP2,*
**N**) in the larval gut of *T. molitor* against microbial challenge with *E. coli* (Ec), *S. aureus* (Sa), and *C. albicans* (Ca) were quantified relative to *L27a* at 24 h post-infection. ‘*’ indicates significant difference between ds*TmPGRP-SA* and ds*EGFP*-treated group (*p* < 0.05); ns = not significant.

**Figure 8 ijms-21-02113-f008:**
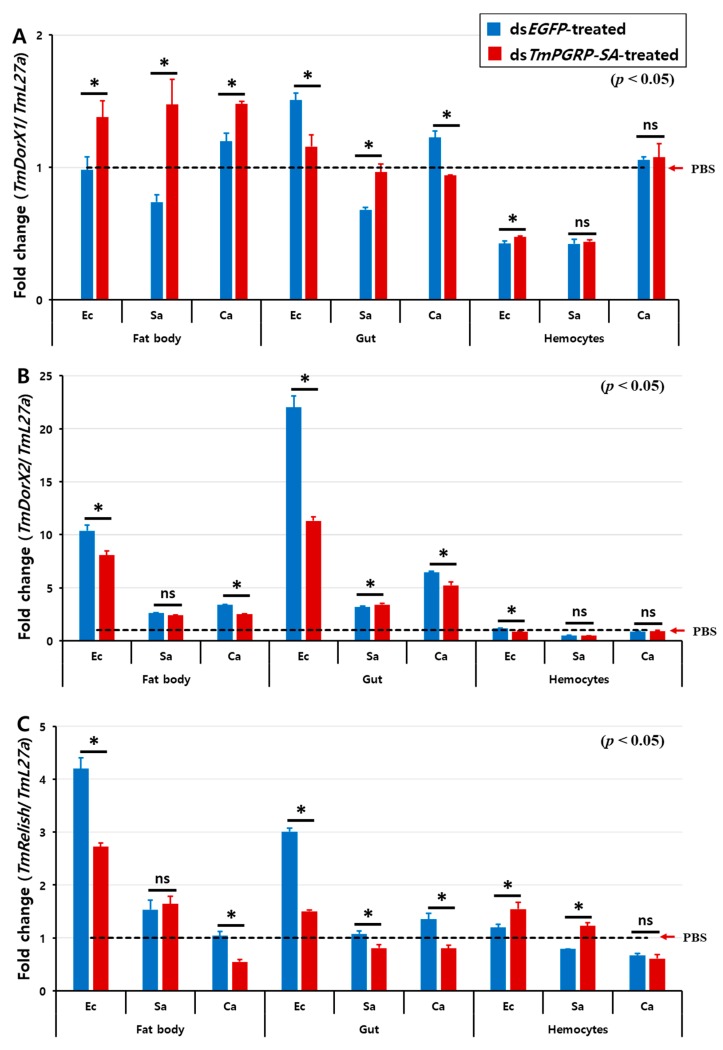
Relative transcript level of three NF-κB transcription factors in *TmPGRP-SA* knockdown insects following microbial infection. The average fold-change of *TmRelish* (**A**), *TmDorX1* (**B**), and *TmDorX2* (**C**) in *TmPGRP-SA*-silenced larvae following challenge with *E. coli* (Ec), *S. aureus* (Sa), and *C. albicans* (Ca) was quantified relative to that of *L27a* at 24 h post-challenge. ds*EGFP* served as a negative control. Bars represent mean ± SE of three independent experiments. ‘*’ indicates significant difference (*p* < 0.05); ns = not significant.

**Figure 9 ijms-21-02113-f009:**
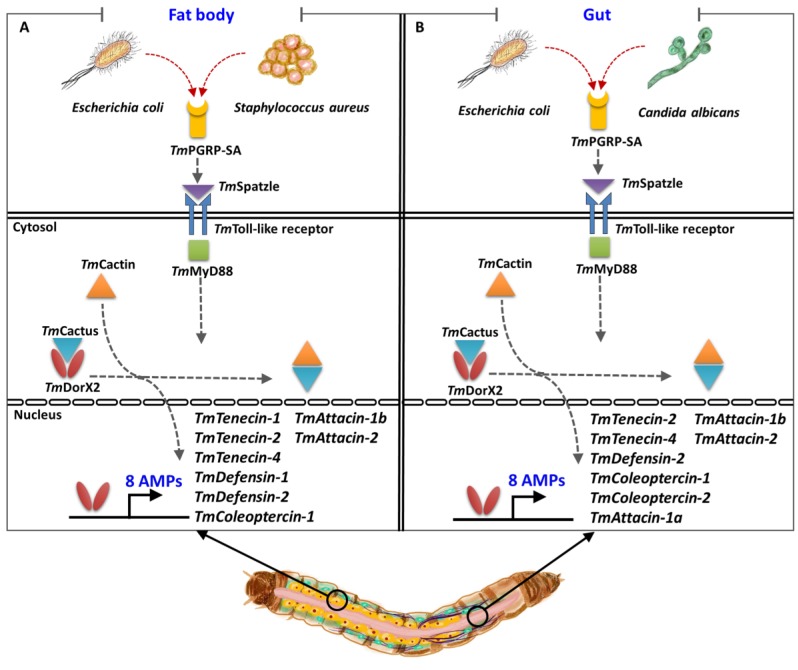
A simplified schematic representation of the key role of *Tm*PGRP-SA in regulating antimicrobial peptide (AMP) expression in the larval fat body (**A**) and gut (**B**) following infection with *E. coli*, *S. aureus*, and *C. albicans*.

**Table 1 ijms-21-02113-t001:** The accession number of PGRP-SA proteins used for bioinformatic analysis of this study.

Name	Abbreviation	GenBank Number
*Tribolium castaneum* PREDICTED: peptidoglycan recognition protein 2	*Tc*PGRP-SA	P_008192927.1
*Locusta migratoria* PGRP-SA	*Lm*PGRP-SA	AFD54029.1
*Bombus consobrinus* peptidoglycan recognition protein SA	*Bc*PGRP-SA	ATL64828.1
*Bombus koreanus* peptidoglycan recognition protein SA	*Bk*PGRP-SA	ATL64813.1
*Gryllus bimaculatus* peptidoglycan recognition protein SA	*Gb*PGRP-SA	BBG28438.1
*Drosophila simulans* PGRP-SA	*Ds*PGRP-SA	XP_002106687.1
*Drosophila busckii* PGRP-SA	*Db*PGRP-SA	ALC48229.1
*Eumeta japonica* Peptidoglycan recognition protein SA	*Ej*PGRP-SA	GBP17419.1
*Drosophila melanogaster* peptidoglycan recognition protein SA	*Dm*PGRP-SA	CAD89124.1
*Bactrocera latifrons* Peptidoglycan recognition protein SA	*Bl*PGRP-SA	JAI23539.1
*Zeugodacus cucurbita* Peptidoglycan recognition protein SA	*Zc*PGRP-SA	JAD13283.1
*Bombus ignites* peptidoglycan recognition protein SA	*Bi*PGRP-SA	ATL64812.1
*Homo sapiens* Peptidoglycan recognition protein 3	*Hs*PGLYRP3	AAI28116.1

**Table 2 ijms-21-02113-t002:** Sequences of the primers used in this study.

Primer	Sequence (5′–3′)
*Tm*PGRP_SA_qPCR_Fw *Tm*PGRP_SA_qPCR_Rv	5′-TCAATGCGGAGTTGAACTGGGAGA-3′ 5′-TAGAGTTTCAGTCCAGGGCTGCTT-3′
ds*Tm*PGRP-SA_Fw ds*Tm*PGRP-SA_Rv	5′-TAATACGACTCACTATAGGG AGAGACTCGGAAAGCGAATGTGC-3′ 5′-TAATACGACTCACTATAGGG AGACTCCCAGTTCAACTCCGCAT-3′
dsEGFP_Fw dsEGFP_Rv	5′-TAATACGACTCACTATAGGG TACGTAAACGGCCACAAGTTC-3′ 5′-TAATACGACTCACTATAGGG TTGCTCAGGTAGTGTTGTCG-3′
*Tm*Tenecin-1_Fw *Tm*Tenecin-1_Rv	5′-CAGCTGAAGAAATCGAACAAGG-3′ 5′-CAGACCCTCTTTCCGTTACAGT-3′
*Tm*Tenecin-2_Fw *Tm*Tenecin-2_Rv	5′-CAGCAAAACGGAGGATGGTC-3′ 5′-CGTTGAAATCGTGATCTTGTCC-3′
*Tm*Tenecin-3_Fw *Tm*Tenecin-3_Rv	5′-GATTTGCTTGATTCTGGTGGTC-3′ 5′-CTGATGGCCTCCTAAATGTCC-3′
*Tm*Tenecin-4_Fw *Tm*Tenecin-4_Rv	5′-GGACATTGAAGATCCAGGAAAG-3′ 5′-CGGTGTTCCTTATGTAGAGCTG-3′
*Tm*Defensin-1_Fw *Tm*Defencin-1_Rv	5′-AAATCGAACAAGGCCAACAC-3′ 5′-GCAAATGCAGACCCTCTTTC-3′
*Tm*Defensin-2_Fw *Tm*Defensin-2_Rv	5′-GGGATGCCTCATGAAGATGTAG-3′ 5′-CCAATGCAAACACATTCGTC-3′
*Tm*Coleoptericin-1_Fw *Tm*Coleoptericin-1_Rv	5′-GGACAGAATGGTGGATGGTC-3′ 5′-CTCCAACATTCCAGGTAGGC-3′
*Tm*Coleoptericin-2_Fw *Tm*Coleoptericin-2_Rv	5′-GGACGGTTCTGATCTTCTTGAT-3′ 5′-CAGCTGTTTGTTTGTTCTCGTC-3′
*Tm*Attacin-1a_Fw *Tm*Attacin-1a_Rv	5′-GAAACGAAATGGAAGGTGGA-3′ 5′-TGCTTCGGCAGACAATACAG-3′
*Tm*Attacin-1b_Fw *Tm*Attacin-1b_Rv	5′-GAGCTGTGAATGCAGGACAA-3′ 5′-CCCTCTGATGAAACCTCCAA-3′
*Tm*Attacin-2_Fw *Tm*Attacin-2_Rv	5′-AACTGGGATATTCGCACGTC-3′ 5′-CCCTCCGAAATGTCTGTTGT-3′
*Tm*Cecropin-2_Fw *Tm*Cecropin-2_Rv	5′-TACTAGCAGCGCCAAAACCT-3′ 5′-CTGGAACATTAGGCGGAGAA-3′
*Tm*Thaumatin-like protein-1_Fw *Tm*Thaumatin-like protein-1_Rv	5′-CTCAAAGGACACGCAGGACT-3′ 5′-ACTTTGAGCTTCTCGGGACA-3′
*Tm*Thaumatin-like protein-2_Fw *Tm*Thaumatin-like protein-2_Rv	5′-CCGTCTGGCTAGGAGTTCTG-3′ 5′-ACTCCTCCAGCTCCGTTACA-3′
*Tm*Relish_qPCR_Fw *Tm*Relish_qPCR_Rv	5′-AGCGTCAAGTTGGAGCAGAT-3′ 5′-GTCCGGACCTCATCAAGTGT-3′
*Tm*DorX1_qPCR_Fw *Tm*DorX1_qPCR_Rv	5′-AGCGTTGAGGTTTCGGTATG-3′ 5′-TCTTTGGTGACGCAAGACAC-3′
*Tm*DorX2_qPCR_Fw *Tm*DorX2_qPCR_Rv	5′-ACACCCCCGAAATCACAAAC-3′ 5′-TTTCAGAGCGCCAGGTTTTG-3′
*Tm*L27a_qPCR_Fw *Tm*L27a_qPCR_Rv	5′-TCATCCTGAAGGCAAAGCTCCAGT-3′ 5′-AGGTTGGTTAGGCAGGCACCTTTA-3′

Underlines indicate T7 promoter sequences.

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
