# Peer review of "TmPGRP-SA regulates Antimicrobial Response to Bacteria and Fungi in the Fat Body and Gut of Tenebrio molitor"

_ijms, 2020, doi:10.3390/ijms21062113_

Round 1

Reviewer 1 Report

Authors studied how the already known peptidoglycan recognition protein-SA (TmPGRP-SA) from the mealworm Tenebrio molitor mediates regulation of antimicrobial response to gram-positive and gram-negative bacteria as well as against fungi infection. The experiments clearly show Tm-PGRP-SA as a positive regulator in fat body and gut (but not in hemocytes), regulating the expression of 8 of 14 antimicrobial peptide (AMP) genes in each of the tissues through the Toll pathway. Similar studies had previously been done on the model insect Drosophila, but are novel for a coleopteran species.

The manuscript is generally well written and needs only minor revision. My only major comment is that it would have been nice to see whether changes in AMP expression in fat body and gut are reflected by changes in AMP titers in the hemolymph.

Minors:

  • line 24: delete "the" regulating...
  • line 48: which are known...
  • line 132: either say "Orthoptera" or "orthopterans"
  • line 179: capitalize "Malpighian" as in following lines
  • Fig. 3: I miss statistics in A and explanation for different lower letters in B and C
  • Lines 217/218: this should be part of Discussion. I recommend to strictly separate Results and Discussion (see also line 243/244
  • Fig. 8: delete the numbers for transcription levels above the bars; this is confusing and values can be read from the Y-axis
  • line 369: amino acids
  • line 516: values for centrifugation steps have to be presented as g-values and not in rpm
  • line 543: authors used 10th to 12th instar larvae, which is not really "young"
  • line 579: "is" a ....
  • References have to be written according to Authors Instructions for "Insects" (species names in italics; correct acronyms for journal titles etc.)

Author Response

Dear Reviewer 1,

Best regards,

Reviewer 2 Report

In vitro and in vivo studies have primarily addressed the role of TmPGRP-SA as an innate immune recognition molecule that initiates the prophenoloxidase (proPO) cascade as well as the Toll signaling pathway. However, the tissue-specific role of TmPGRP-SA in inhibiting the Toll pathway needs to be elucidated. In this study, the authors sought to understand the functional role of extracellular TmPGRP-SA in the survival and AMP gene expression of T. molitor larvae in response to infections with E. coli, S. aureus, and C. albicans by utilizing an RNA interference (RNAi) approach. They also analyzed the expression pattern of NF-κB genes in T. molitor larvae following TmPGRP-SA silencing and infection.

The introduction was a delight to read and is a thorough summation of the state of the field as it relates to PGN recognition. The experimental design is sound and the results are relatively easy to interpret. I have only minor comments / suggestions for improving the manuscript with regard to data presentation and interpretation, which should be relatively easy for the author's to address.

Minor comments:
Line 93-94: I am uncertain whether it is proper to say they have been characterized without providing sufficient reference, particularly as a large number of ligands are mentioned (editor's discretion).

Figure 1-2: Highlighting the active site residues and the sequence alignments are helpful. Have the authors considered running their predicted amino acid sequence through a structural prediction algorithm and mapping the predicted structure onto one of the solved structures for PGRP-SA? It would allow them to better illustrate/visualize the conserved nature (or not) of the residues of the substrate-binding site.

Figure 3: Do the authors have a hypothesis as to why fat body expression of PGRP-SA is so highly prevalent in larva as compared to the integument? Does this hint at barrier deficiencies due to rapid growth / turnover of the integument?

Figure 3-4: What to make of the absence of PGRP-SA in the gut? Is this the case for other insects as well? Are their hypotheses for why it displays this particular tissue tropism? Is the microbiome known and are the ratios of gram positive to negative commensals known (could explain differential reaction of PGRP-SA to E. coli in Figure 4). Additionally, regarding the control for normalizing the RT-qPCR, what is known about the baseline rates of protein synthesis during the developmental cycle of T. molitor and would differences significantly affect result interpretation? Additionally, how does infection affect rates of protein synthesis and expression of TmL27a?

Figure 4: Larval fat body expression is reported to be high in Figure 3. How do the author's explain the more rapid expression during C. albicans infection …? Perhaps standardizing the axis for fold change between graphs would lessen the impact of panel C, as at present, the reader will question why C. albicans infection results in such a high increase, when in reality it is similar to that of S. aureus … which is also confusing!

Figure 5: What do the authors think of the potential that death occurs simply from the outgrowth of either commensals or other, opportunistic pathogens invading concurrently in their model system? This would potentially explain the effects of the knockdown of TMPGRP-SA on C. albicans, which does not synthesize PGN. It would also potentially explain differences in survival between Sa and Ec infections. Plating / cfu counts of the dead larva (assuming cultivability) would establish whether other bacterial commensals / opportunistic pathogens are responsible for enhanced death.

Line 309-310, 331-333: Again, the concern is that rather than responding directly to C. albicans, these observations may indicate that the commensal flora/fauna of the gut may achieve dysbiosis during C. albicans infection. In that respect, TmPGRP-SA would be responding not to C. albicans, but to the dysbiosis caused by C. albicans and potential infiltration of and overgrowth by commensal biota. Even without overgrowth, the damage caused by C. albicans may simply enhance the sensitivity of the gut to the presence of the microbes that normally reside there. Przemieniecki et al. (2020) appear to show that the microbiome of T. molitor skews towards the Gram negatives (Proteobacteria, Bacteroidetes). In rationalizing why differences exist between the fat body and gut as it relates to TmPGRP-SA and C. albicans infection, the logical answer is that the gut microbiome is having a discernible, combinatorial effect. Highly controlled studies utilizing germ-free / antibiotic-treated T. molitor would be needed to access this further.

Author Response

Dear Reviewer 2,

Best regards,

Round 2

Reviewer 1 Report

Authors have improved their manuscript according to my previous suggestions. I understand that measuring AMP contents in the hemolymph will be done in future.

Reviewer 2 Report

The authors have sufficiently addressed the previously stated minor comments.

A new minor concern has arisen as a result of one of their responses:

I highly advise that the author's include their T. molitor rearing protocol, emphasizing the use of antibiotics / antifungals as this information is also essential for the accurate interpretation of their results. As such, it should be presented in the methods section in full, as opposed to referencing a previous article. Rearing any specimen under germ-free conditions will directly influence the development of PAMP recognition systems. 

"... eggs were collected and were fed on the artificial diet consisting of 170 g wheat flour, 0.5 g chloramphenicol, 20 g roasted soy flour, 0.5 g sorbic acid, 0.5 mL propionic acid, 10 g soy protein, and 100 g wheat bran in 200 mL of distilled water, autoclaved at 121 °C for 20 min. Therefore, the experimental units have been reared using antibiotic (chloramphenicol) and antifungal (propionic acid)."

The authors should also include when during the course of their experimentation the antibiotic / antifungal is removed from use.

Author Response

Sincerest thanks for your response and comments on our manuscript. We have revised the rearing stock of T. molitor as follows:

Young larvae were fed an artificial diet consisting of 170 g wheat flour, 0.5 g chloramphenicol, 20 g roasted soy flour, 0.5 g sorbic acid, 0.5 mL propionic acid, 10 g soy protein, and 100 g wheat bran in 200 mL of distilled water, autoclaved at 121 °C for 20 min. Healthy and fed 10th –12th instar T. molitor larvae (approximately 2.4 cm) were used for all the experiments. The experimental units maintained in an insectary with artificial diet during all experiments.